# A Transformation of the Approach to Evaluating a Region's Investment Attractiveness as a Consequence of the COVID-19 Pandemic

**Dmitriy G. Rodionov, Evgenii A. Konnikov * and Magomedgusen N. Nasrutdinov**

Graduate School of Industrial Economics, Peter the Great St. Petersburg Polytechnic University,
195251 St. Petersburg, Russia; rodion_dm@mail.ru (D.G.R.); magomedgusen.nasrutdinov@mail.ru (M.N.N.)
* Correspondence: konnikov.evgeniy@gmail.com; Tel.: +7-961-808-4582

**Abstract:** The global COVID-19 pandemic has caused a transformation of virtually all aspects of the world order today. Due to the introduction of the world quarantine, a considerable share of professional communications has been transformed into a format of distance interaction. As a result, the specific weight of traditional components of the investment attractiveness of a region is steadily going down, because modern business can be built without the need for territorial unity. It should be stated that now the criteria according to which investors decide if they are ready to invest in a region are dynamically transforming. The significance of the following characteristics is increasingly growing: the sustainable development of a region, qualities of the social environment, and consistency of the social infrastructure. Thus, the approaches to evaluating the region's investment attractiveness must be transformed. Moreover, the investment process at the federal level involves the determination of target areas of regional development. Despite the universal significance of innovative development, the region can develop much more dynamically when a complex external environment is formed that complements its development model. Interregional interaction, as well as an integrated approach to innovative development, taking into account not only the momentary effect, but also the qualitative long-term transformation of the region, will significantly increase the return on investment. At the same time, the currently existing methods for assessing the investment attractiveness of the region are usually heuristic in nature and are not universal. The heuristic nature of the existing methods does not allow to completely abstract from the subjectivity of the researcher. Moreover, the existing methods do not take into account the cyclical properties of the innovative development of the region, which lead to the formation of a long-term effect from the transformation of the regional environment. This study is aimed at forming a comprehensive methodology that can be used to evaluate the investment attractiveness of a certain region and conclude about the lines of business that should be developed in it as well as to find ways to increase the region's investment attractiveness. According to the results of the study, a comprehensive methodology was formed to evaluate the region's investment attractiveness. It consists of three key indicators, namely, the level of the region's investment attractiveness, the projected level of the region's investment attractiveness, and the development vector of the region's investment attractiveness. This methodology is based on a set of indicators that consider the status of the economic and social environment of the region, as well as the status of the innovative and ecological environment. The methodology can be used to make multi-dimensional conclusions both about the growth areas responsible for increasing the region's innovative attractiveness and the lines of business that should be developed in the region.

**Keywords:** region's investment attractiveness; economic environment; social environment; innovative environment; ecological environment; fuzzy logic; regARIMA; combined effect



## 1. Introduction

The integration of digital communications tools into the operations of enterprises is growing exponentially today. The world COVID-19 pandemic has intensified these

processes a lot due to the compulsory isolation of a considerable part of labor resources. This involuntary integration of digital technology into virtually all business processes of enterprises can eventually end up in a complete transformation of the world economy and reduce the significance of the territorial belonging of labor force. This trend can decrease the role of the maturity of regional business structures, territorial belonging of educational institutions, and the overall complexity of the regional infrastructure and resource base. Being able to work from virtually any point of the world, people will no longer be dependent on the regional specifics (Matteucci 2015). No doubt, this trend is not new, but it is the world COVID-19 pandemic that has provoked its active spread. The consequence of this trend is a mass reduction in the investment attractiveness of regional entities. Since there is no need to interact physically to carry out business processes, enterprises are losing motivation for developing the regional infrastructure, which, in turn, reduce their investment activity. Thus, the tools for managing the region's investment attractiveness are becoming more and more important. One of the main stages in the management process is valuation, which, in case of the region's investment attractiveness, becomes especially complex in the context of the above trends. Since regions are getting less and less interesting for investment at a physical level, other regional characteristics, which were not clearly comprehended before, are becoming especially important, such as: the sustainable development of the environment, the ecological status of the environment and the specifics of social communication, the integration of the life-sustaining systems and the level of their digitalization, etc. (Casi and Resmini 2017). Regions with the well-developed social and business infrastructure are becoming more and more attractive for investment (Vilken et al. 2019).

According to the above, it should be noted that there is need for principally new tools for assessing the level of regions' investment attractiveness. These tools must consider the complexity and diversity of the region in the context of modern realities plus evaluate its development trends. The existing methods for assessing the investment attractiveness of objects are primarily focused on assessing the investment attractiveness of an enterprise. The application of these methods to assess the investment attractiveness of the region is ineffective, since the region, as an object of integrated development, cannot be focused solely on the increment of economic results. Being a more complex object, the region requires taking into account the state of all subsystems, such as the social environment or the ecological environment. At the same time, the potential return on investment in the region is not instantaneous, since any one-dimensional development presupposes a comprehensive transformation of the entire regional environment. Consequently, the methodology for assessing the investment attractiveness of a region should take into account the dynamic relationship between indicators of regional development. It should be also noted that the existing methods for assessing the investment attractiveness of a region are usually based on a heuristic methodology and are not universal. A modern effective methodology for assessing the investment attractiveness of a region should be based on universal quantitative parameters, as well as on scientifically proven criteria for describing the relationship between quantitative variables. The purpose of this study is to form a comprehensive methodology that can be used to evaluate the investment attractiveness of a certain region and, as a result, make conclusions about the lines of business that should be developed in it as well as to find ways to increase the region's investment attractiveness. In order to reach this goal, the following objectives have to be achieved:

1. Critical study of the theoretical basis in the field of the region's investment attractiveness evaluation;
2. Formation of the comprehensive characteristics of the region attractive for investment in the context of the current development trends in the world;
3. Development of a conceptual methodology for evaluating the region's investment attractiveness that contains a set of valuation tools and a procedure for their application. The methodology should also regulate the specifics of multi-level conclusion-making depending on the subject of valuation;

4. Development of a set of tools for evaluating the components of the region's investment attractiveness;
5. Trial of the developed set of tools for evaluating the components of the region's investment attractiveness and making multi-level conclusions based on the results of valuation.

According to these objectives, the main outcome of this research is a new comprehensive methodology for evaluating the region's investment attractiveness that corresponds to today's digitalization trends.

## 2. Literature Review

The region's investment attractiveness is a complex concept (Snieska and Zykiene 2015). Modern science lacks a consensus about the essence and structural integrity of investment attractiveness of such an entity as a region. However, several central factors of the region's investment attractiveness can be highlighted. Quite a large part of the scientific community defines the level of accessibility of regional resources as a key indicator of the region's investment attractiveness in addition to the ability of regional authorities to update and enlarge the resource base comparing to other regions (competing regions) (Snieska and Zykiene 2015). According to this statement, the region's investment attractiveness is looked at as a comparative characteristic, while the criteria for comparison are based on the specifics of the region's resourcing. In particular, the study of investment attractiveness of the Lithuanian city of Alytus carried out in 2015 shows that it is the value of resources and the level of market competition that define the level of the indicator being considered (Snieska and Zykiene 2015). However, the level of market competition at the regional level determines exclusively the intraregional dynamics of development, while investment attractiveness is a comparative interregional characteristic. Also, significant discussions are caused by the differentiation of approaches to determining the resource base for the development of the region. Various scientists differentiate the specific weight of the impact made by various types of resources on the region's investment attractiveness. In the above study of Alytus, the authors emphasize the priority of qualification and value of labor resources (Snieska and Zykiene 2015). This statement is scientifically substantiated, since it is the availability of qualified labor force that defines the innovative development of the region, which, in turn, is one of the region's key competitive advantages (Bruneckienė et al. 2012). At the same time, it is also necessary to remember about the interregional and international differentiation of approaches to determining the level of qualifications, as well as the differentiation of innovative specialization of regions. This fact raises the question of the comparability of the characteristics of the labor force skill level. Other scientists believe that the differentiation of the specific weight of various types of resources is not necessary and that resources should be considered in a totality, in particular, labor, transport, material, and natural resources (Glebova et al. 2015; Kinda 2013). The statement about the need for competitive interaction between regions is also solid. Regional competition stimulates innovative development and increases labor productivity, and thus creates value added (Stern 2002). At the same time, the development of a competitive environment is dialectical in relation to regional development in general, since one of the results of industrialization can be environmental damage, and the consequence of the automation of large industrial enterprises in the region can be an increase in unemployment. Thus, the innovative development of the region should be considered strategically, taking into account the dynamic transformation of the entire environment of the region (social, economic, etc.).

Other members of the scientific community define the sophistication of the regional infrastructure as the basis for developing the region's investment attractiveness rather than the quality of resourcing (Windhyastiti et al. 2019).

It is highlighted that the need for investing in the transport infrastructure is a major infrastructural characteristic (Hallward-Driemeier et al. 2006). The priority of infrastructure as a basis for developing the region's investment attractiveness is also substantiated by a growth in labor productivity and, consequently, by a growth in the region's competitiveness.

At the same time, the consequence of infrastructure development can also be environmental transformation and a potential increase in the level of competition in the labor market, which in turn can lead to social stratification. This thesis also confirms the dialectic of the region's innovative development. Regional autonomy can act as a mediator for the development of infrastructure. It should be noted that the development of regional infrastructure is the effect of multiple factors and there are quite a few ways to stimulate this interaction. A possible example of this is the process of transferring the key regional management and social infrastructure from production centers to the areas of creative activity and consumption (Hallward-Driemeier et al. 2003).

Some representatives of the scientific community claim that the basis for developing the region's investment attractiveness is to make sure that administrative and regulatory barriers are comparatively low (primarily, the level of tax burden), and to create special economic zones (Marona et al. 2012). At the same time, it must be remembered that this issue is fully or partially controlled at the federal, and not at the regional level. Consequently, within the framework of building a strategy for innovative development, the region is limited from the point of view of managing this specificity. The development of special economic zones also presupposes a unique factor specificity. Despite the worldwide recognition of the effectiveness of this tool in increasing the investment attractiveness of the region, the systematic nature of the interaction of regional resources remains the primary factor (Nazarczuk and Cicha-Nazarczuk 2021). Some scientists suggest that the maturity of the regional legislation and the guarantees of its effectiveness encourage economic growth, which, in turn, helps to attract additional investments. This theory highlights one of the major qualities of the region's investment attractiveness—its cyclic nature. The attracted regional investments are used to develop the region, which can be expressed both in the sophistication of the regional infrastructure and in the development of the quality of resourcing, resulting in the region's bigger investment attractiveness. This statement is also consistent with the claim that regional investments are bound to affect the economic results of regional business, which, in turn, leads to an increase in the value of regional resources (Batik 2013). The repeating pattern of investment attractiveness defines its dynamic nature, which is expressed in its versatility, being the effect of intra-regional changes and the development of competing regions (Michalet 1999).

Nevertheless, the overwhelming majority of the scientific community claims that the region's investment attractiveness is expressed in the unity of the above factors. This statement is supported by many empiric research studies. The study of the investment attractiveness of some regions in Romania in 2014 showed that the basic factors of its development are: (1) the quality and value of labor resources; (2) the specifics of agglomeration; (3) the maturity of the infrastructure; (4) the quality of the intellectual basis; (5) the maturity of the market; (6) the comparative value of resources (Danciu and Strat 2014). This research, in a quantitative way, substantiates the multiplicative impact of these factors on the region's investment attractiveness. The overwhelming majority of these indicators are qualitative, and the methodology for calculating them can vary significantly depending on the country and region. This fact does not allow for speaking about the possibility of extrapolating the results of this study, however, a detailed analysis of their content allows to put forward key components that universally determine the innovative development of the region. The study "What investors want: A guide for cities" showed that the region's investment attractiveness depends on: (1) mature regional economy that has a room for growth; (2) presence of highly-qualified labor force; (3) resilience to economic collapses and external impact; (4) well-developed transport infrastructure, both at the regional and international levels; (5) the regional administration, encouraging investments in the region, having a high authority at the national level and implementing consistent policy of the region's development; (6) a flexible investment system and readiness of the administration to facilitate investments (McDonald and Bailly 2017). Thus, the initial point is confirmed. According to it, investment attractiveness is a complex category that manifests itself in multiplicative interaction of a totality of factors. Consequently, the region's investment

attractiveness cannot be studied outside the context of the region's development as a multidimensional environment. However, the factors formulated in the framework of this study are also extremely non-universal, and can be differentiated from the point of view of calculation methodology.

One of the most effective areas for the growth of the region's investment attractiveness is its specialization, which, as a rule, is expressed in the development of regional clusters (Borkova et al. 2019; Kudryavtseva and Skhvediani 2020). Cluster strategies have been used in the developed countries for a long while to increase the investment attractiveness of territories (Xu et al. 2008). Regional clusters make it possible to exponentially reduce the potential expenses of regional industrial enterprises due to better logistic processes, information processes, etc. They help the region to considerably increase its innovative potential and create resources for higher competitiveness of business due to the contribution of research and development. Clusters also formalize the vector of the region's development, which makes the risks faced by all participants in the economic system go down (Mitrofanova and Korepova 2015; Morschett et al. 2015). The territory of many countries (including Russia) is significantly differentiated in terms of resources, climate, infrastructure, culture, and demography. Regions can have limited infrastructure, but be rich in natural resources, which inevitably leads to the creation of industrial clusters, while the boundary and climate specifics of other regions encourage trade, logistic, marketing and consulting specialization (Tao and Mengqi 2019). Thus, it is full or partial use of cluster approach to creating the development strategies of the region that makes it possible to dynamically and effectively increase its investment attractiveness. The basic hypothesis of this study is that the investment attractiveness of the region is a complex parameter, the increase of which implies modeling the return on management of the region's environment. On the basis of this hypothesis, hypotheses-consequences can be formulated, which are subdivided into instrumental hypothesis and structural hypothesis. According to the instrumental hypothesis, the investment attractiveness of the region can be described as a relative value reflecting the one-time state of the region's environment quantifiers, taking into account the mutual influence of these variables. According to the structural hypothesis, the region's environment can be hierarchically differentiated into a set of sub-environments, the state of each of them can be represented by a set of quantifier indicators. Detailing and confirmation of the instrumental and structural hypotheses is necessary for confirming the basic hypothesis.

Thus, the region's investment attractiveness is formed under the effect of many factors, such as the availability of natural resources, macroeconomic conditions, the availability of qualified labor force and its territorial distribution, the administrative and information barriers to entering the market, the condition of the competitive environment, tax legislation (in particular, tax incentives), the development level of the financial market, the legislative security of investors, inflation stability, political stability and the social security of the population, the quality of state governance, the crime and corruption rate, the openness of the economy in terms of trade, as well as the maturity of the objects of infrastructure, their accessibility, and uniform distribution in the territory. In accordance with the specified totality of factors, the investment attractiveness of the region as a cluster is defined by a comparative status of the business environment expressed in the potential return on investment, the level of economic and political risks, innovative potential (in case of innovative business), the consistency and stability of economic relations, as well as the comparative level of integration and development costs and the state of the non-entrepreneurial infrastructure, which implies a comparatively high standard of living, a well-developed educational and health care environment, a sufficient level of ecological requirements and the lack of cultural and ethnical obstacles to business activities. Thus, the region's investment attractiveness can be expressed by the state of two integral blocks, such as, the status of the business environment, differentiated for economic environment and innovative environment, and the status of the non-entrepreneurial environment, differentiated for social environment and ecological environment (Figure 1).

**Figure 1.** The elementary model of the region's investment attractiveness.

### 3. Materials and Methods

In accordance with the elementary model of the region's investment attractiveness, formed as a result of theoretical research, 4 key aspects of its formations or 4 environments are pointed out. Each of the environments is complex and its state can be expressed by many indicators. Let us first look at the economic environment. This environment reflects the ability of business to effectively manage costs by return on investment, the status of the competitive environment, the level of currency and political risks, as well as other properties of the region's investment climate that have a direct effect on the key business processes and activities of the investor inside the region. One of the primary consequences of the development of this environment is the increment of the number of profitable regional enterprises (Tao and Mengqi 2019). This indicator directly reflects the probabilistic potential return on investment, which is a primary property of the region with high investment attractiveness. In addition, as a result of the increased aggregated outcome achieved due to the economic activities of the region's enterprises and due to the development level of financial institutions, the turnover of regional enterprises and organizations grows, which, in turn, indirectly indicates the amount of monetary funds in the regional economy (Tceberko et al. 2018). This indicator is important for the investor, because it also reflects the potential increment of effective demand and the relative accessibility of credit resources, as well as the volumes of incoming investments, which can be the good evidence of low investment risks. The dual indicator of the investment attractiveness of the region's economic environment can be the price index of goods and services in the region (McCully et al. 2007). On the one hand, inflation processes are a sign of high currency risk. However, they can be the result of social and economic development. Thus, the relative stability of this indicator can confirm the economic sustainability of the region, and, therefore, its investment attractiveness. In addition, due to a growth in the aggregate result of enterprises' business operations in the region, export activity, which is indicated by the volume of goods shipped for export in the region, increases (Hellman 2006). An increase in this indicator shows that the potential return on investment is growing. The effect of developing insurance and financial institutions is manifested in the fact that the number of voluntary insurance payments is growing and the loan and credit debt of enterprises is going down. A reduction in these indicators is the sign of a healthier regional economic environment, which improves the region's investment attractiveness.

At this point, it is essential to consider the following constituent of the business environment—innovative environment. In this case the resulting indicators can be the expenses of the region's enterprises related to technological innovations and the number of advanced production technologies developed in the region (Selentyeva et al. 2018; Schepinin et al. 2018). The expenses related to technological innovations are the first order

consequence and they essentially act as a retransmission indicator. In this case the resulting indicator of an increase in the region's investment attractiveness is the increment of the number of the advanced production technologies that have been created (Kuzovleva et al. 2019).

The same important constituent of the region's investment attractiveness is the well-developed infrastructure, differentiated into social and ecological environment. In case of social environment, the result of positive transformation is the reduced level of unemployment, growth in expenses related to labor remuneration in regional enterprises and organizations (Theodore 2017). The effect of the well-developed social environment is a rise in the price index on the regional housing market, which largely increases the investment attractiveness of the regional construction sector. At the same time, the development of the ecological environment is much more inconsistent in terms of structure. The indicators of the developed ecological environment can be the morbidity rate together with the capacity of the regional health care institutions. The trends in these indicators are consistently related and in case they change negatively, the investment risks related to the development of human capital and management of human resources rise considerably (Rodionov et al. 2018). In addition, from the perspective of long-term investing, a considerable role can be played by the regional expenses arising from land reclamation, since land resources are conditionally non-renewable. Thus, it is possible to present a model for evaluating the region's investment attractiveness consisting of 15 indicators that have been highlighted (Figure 2).

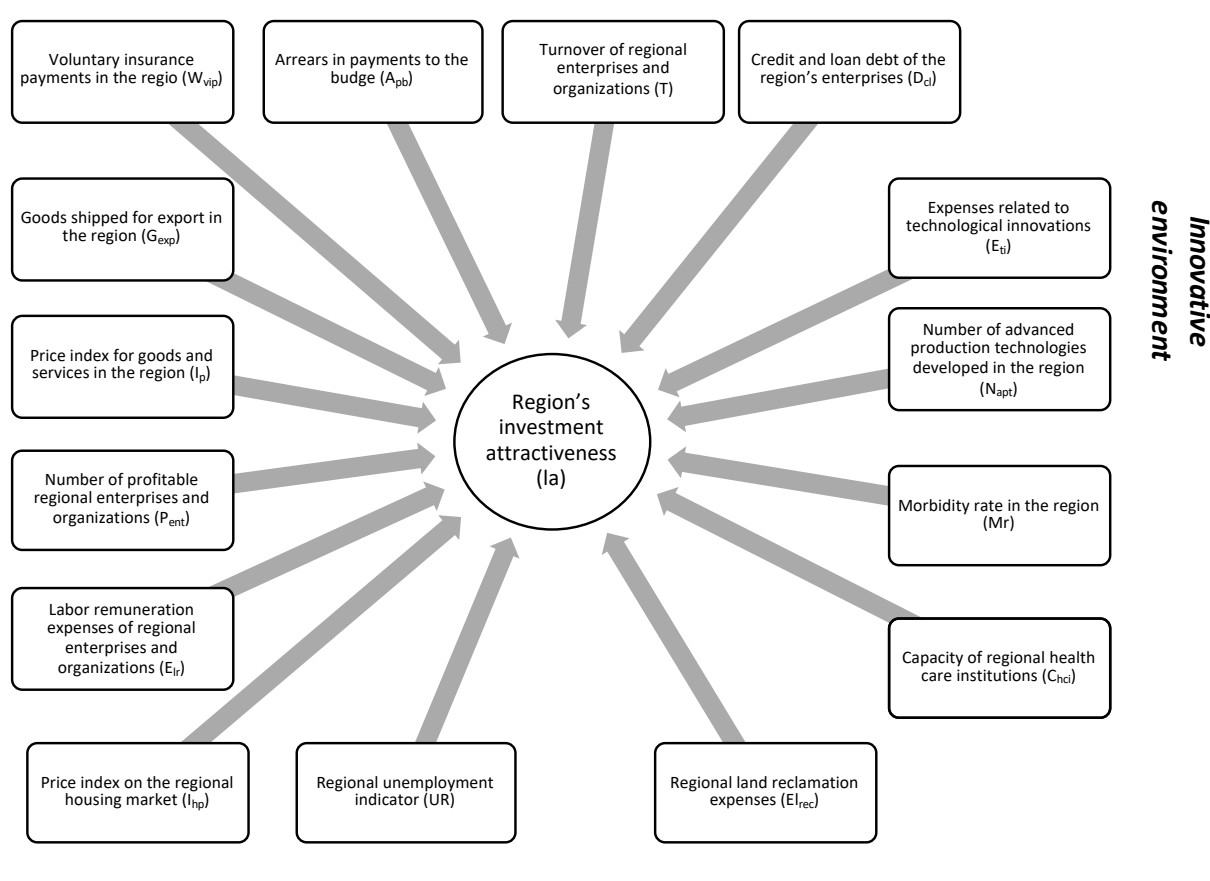

**Figure 2.** The indicative model of the region's investment attractiveness.

The impact of the selected indicators on the region's investment attractiveness cannot be evaluated based on the classical valuation methods. It is largely due to the differentiated nature of the impact created by these indicators, as well as the specifics of the dimension. Moreover, the complexity of the object determines the need to distinguish fuzzy intervals of assessment, which are also characterized by the level of confidence of the expert in his conclusions. Therefore, one of the most appropriate for these purposes is the fuzzy-multiple approach. Methods based on the theory of fuzzy sets are formed by a system of expert assessments, however, unlike statistical and expert assessment methods, they allow to take into account the level of uncertainty by using the membership functions ($\mu(x) \in [0; 1]$) of a subset of a given set. The founder of the application of the theory of fuzzy sets to describe the processes of economic nature is Doctor of Economics, Alexei O. Nedosekin (Rytova and Gutman 2019; Nedosekin et al. 2017), who proposed an algorithm for assessing a complex economic indicator using the theory of fuzzy sets.

As mentioned above, the region's investment attractiveness is a dynamic characteristic. These specifics generate the need to take into account the regional development trends, which is possible if the suggested level of the region's investment attractiveness is forecasted. In this case the most effective way is to project the elements of the suggested fuzzy-multiple model. It is proposed that the regARIMA class of models should be used as tools, since these models can consider the system effect to the highest degree possible, which is crucial for such complex research objects. The built models will be a consistent forecast system of econometric equations.

The projected values, obtained through a reduced set of equations herein after, are aggregated according to the previously described fuzzy-multiple model. By comparing the current and projected values of the integral indicator, the development vector of the region's investment attractiveness can be defined. Thus, the methodology for evaluating the region's investment attractiveness can be represented through the following conceptual model (Figure 3):

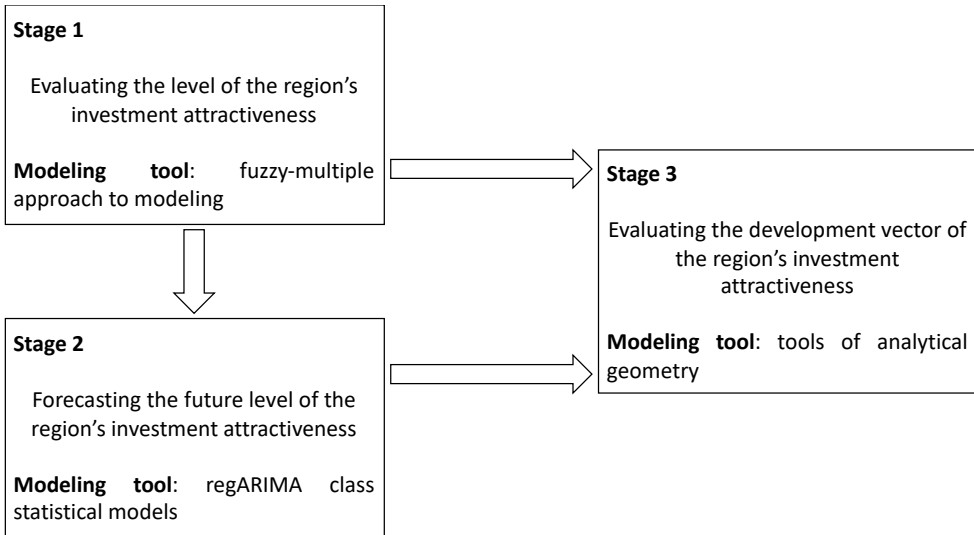

**Figure 3.** Methodology for assessing the region's investment attractiveness.

## 4. Results

The first stage of the proposed methodology is the evaluation of the current level of the region's investment attractiveness. In order to build the fuzzy-multiple model, first of all, it is essential to define the significance of the impact of each of the highlighted indicators. The specific weight of each indicator can vary depending on the regional specifics. So, in order to make the model versatile, it is suggested that the specific weight of the indicators be distributed evenly inside the environment groups, while the specific weight among the

environments should be distributed according to Fishburn's distribution, which implies ranking (Figure 4).

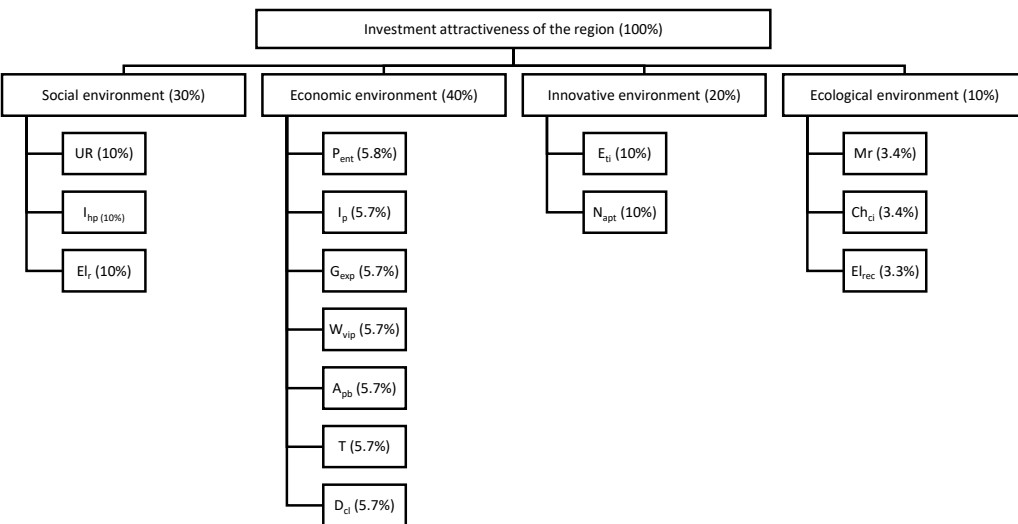

**Figure 4.** Distribution of the specific weight of the indicators affecting the region's investment attractiveness.

These indicators need linguistic variables. One integral linguistic variable is defined in terms of this tool: "The level of the region's investment attractiveness" (I). This term set has 5 subsets:

1. An extremely low level of the region's investment attractiveness;
2. A low level of the region's investment attractiveness;
3. An average level of the region's investment attractiveness;
4. A high level of the region's investment attractiveness;
5. An extremely high level of the region's investment attractiveness.

A linguistic variable has to be formulated for each of the indicators, too. This model consists of 16 indicators. The indicators, despite being heterogeneous and having diverse influence on the integral indicator, can be universalized with one linguistic variable: "Indicator level". This term set has 5 subsets:

1. An extremely high level of the indicator;
2. A high level of the indicator;
3. An average level of the indicator;
4. A low level of the indicator;
5. An extremely low level of the indicator.

The impact of each of the selected linguistic variables is recognized by the means of a specific fuzzy-multiple classifier. A five-level 01-classifier is used for the integral indicator. A segment of the real axis [0; 1] (01-medium) is used as a carrier of the linguistic variable in the classifier. The segment is universal, because any segment of the real axis can be reduced to the segment [0; 1]. A system of five membership functions has been introduced to describe the type of the subsets in the term set. This system characterizes the membership degree of the segment of the 01-carrier values in a given subset (Figure 5).

The graph shows trapezoidal membership functions, where the *Y*-axis indicates the values of the membership functions (from 0 to 1), and the *X*-axis represents terms. Creating a system of fuzzy subsets implies that a set of nodal points is introduced. These modal points are the abscissas of the midpoints of the upper bases of the trapeziums of the classifier. Fuzzy-multiple classifiers are also formed for each particular indicator.

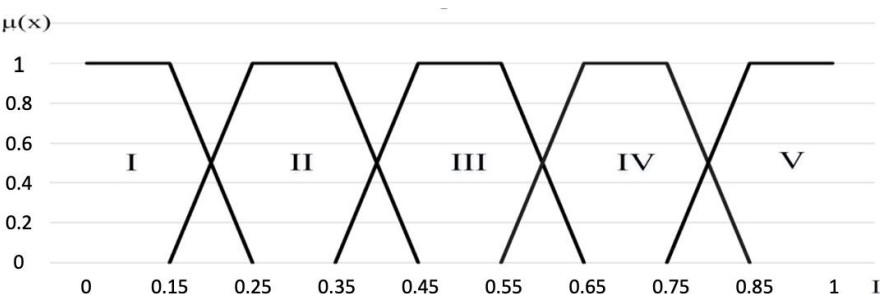

**Figure 5.** The fuzzy-multiple classifier of the values of the region's investment attractiveness level (I).

According to the calculated results of all indicators, their values are recognized by the criterion $\lambda_{iJ} \in [0; 1]$. This indicator correlates the values of the particular indicators with the values of the 01-carrier:

$$\lambda_{iJ} = 1 - \frac{X_i - a_3^*}{a_4^* - a_3^*} \tag{1}$$

where $a_3^*$ and $a_4^*$ are the T-figures of the subset of the term set.

Based on the recognized values of the particular indicators, an integral indicator is calculated:

$$I = \sum_1^{16} p_j \times \sum v_i \times \lambda_{iJ} \tag{2}$$

where

$v_i$ is the weight of the particular indicator, and

$p_j$ are the nodal points of the 01-carrier:

$$p_j = 0.9 - 0.2 \times (j - 1) \tag{3}$$

where $j$ is the number of subsets of the base term set.

Thus, the resulting assessment is determined as weighted average for all the indicators used in valuation, on the one hand, and for all the quality levels of these indicators, on the other hand.

The second stage of the methodology implies forecasting the future values of the indicative figures of the model. The generated models constitute a single forecasting system of econometric equations. To make the presentation more consistent, this system is disaggregated into subsystems, according to the environment being described. The array of primary information on the basis of which the calculations were made is located in the Appendix A. Let us consider the social environment first. This environment is represented by three indicators. Their forecasting models are given in system 4.

$$UR^{n+1} = UR^n - 8.38 - 0.59 * UR^n + 0.98 * MA_{UR^n} - 0.003 * Edu_{bve}^n + 0.0001 * P_{ent}^n$$

$$I_{hp}^{n+1} = I_{hp}^n + 104.95 + 0.64 * I_{hp}^n - 0.9 * I_{hp}^{n-1} + 0.35 * I_{hp}^{n-2} - 0.0001 * P_{ent}^n - 0.86 * S_a^n + 0.009 * S_a^{n2}$$

$$E_{lr}^{n+1} = E_{lr}^n + 2\,869\,691\,800 - 29\,273 * P_{ent}^n - 0.73 * A_{pb}^n \tag{4}$$

where

- $UR^{n+1}$ is the indicator of the regional unemployment rate in period $n + 1$;
- $UR^n$ is the indicator of the regional unemployment rate in period $n$;
- $MA_{UR^n}$ is the error of the predicted UR in period $n$;
- $Edu_{bve}^n$ is the number of state educational organizations of basic vocational education in period $n$;
- $P_{ent}^n$ is the number of profitable regional enterprises and organizations in period $n$;
- $I_{hp}^{n+1}$ is the index of housing prices on the regional market in period $n + 1$;
- $S_a^n$ is the real average monthly salary of the employees working in regional enterprises.

Further, the principle based on which the variables are formed does not undergo any changes, so the equations can be interpreted in accordance with Figure 2. Therefore, no specification is presented further. According to the F criterion and the p-level of the variables, all three models are significant and suitable for forecast, having a confidence level of 95%. The coefficient of determination of the first model is 87.6%, which suggests a sufficient level of dispersion of the dependent variable described by the dispersion of the independent variables. The average error rate is 4.26%, indicating that the forecast is potentially accurate. The coefficient of determination of the second model is 87.2%, while the average percentage of error is 2.1%, which also indicates that the approximation is high quality and the level of predictive power is high. The coefficient of determination of the third model is 99.1%, which is largely due to a stable trend. The average error rate is 2.88%. Figure 6 shows graphs of the actual and theoretical dynamics of the variables.

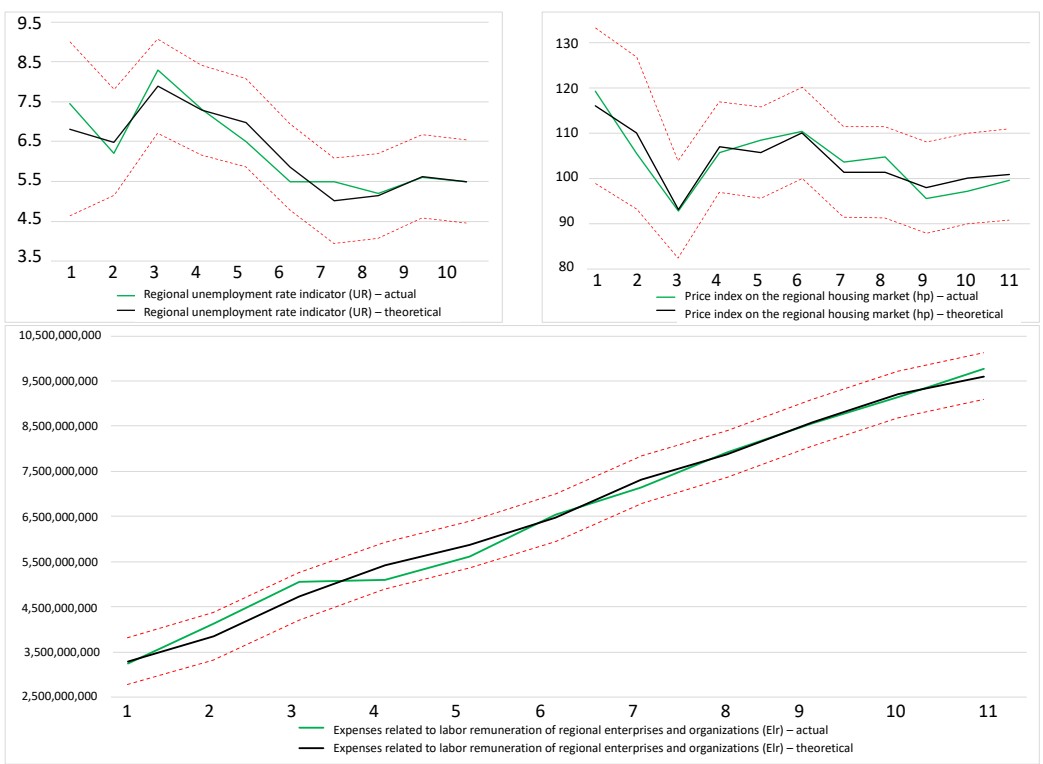

**Figure 6.** Dynamics of actual and theoretical values of the indicators of the social environment.

As you can see, there are no significant structural breaks and structural spikes on the graphs. Thus, the formed system of equations is effective and can be used to forecast the indicators of the region's social environment with a lead period equal to 1 year. However, it should be noted that the constants have to be adjusted according to the specifics of each individual region. It is also worth highlighting that forecasting for a period of more than 1 year ahead is irrelevant, since these projections will be based on theoretical values.

Next, the economic environment of the region should be considered. This environment is the most extensive and represented by 7 indicators. The equations used to forecast these indicators are aggregated into system 5.

$$P_{ent}^{n+1} = 92\,069.65 + 0.55 * P_{ent}^{n} - 0.61 * P_{ent}^{n-1} - 43.59 * N_{apt}^{n}$$

$$I_{p}^{n+1} = 112.24 + 0.4 * I_{p}^{n} - 0.56 * I_{p}^{n-1} - (4.830\text{E} - 11) * T^{n}$$

$$G_{exp}^{n+1} = 10^{\log G_{exp}^{n} + 4.23 - 0.87 * \log G_{exp}^{n} - 0.6 * \log P_{ent}^{n} - 0.44 * \log N_{apt}^{n}}$$

$$W_{dcp}^{n+1} = 10^{\log W_{vip}^{n} + 2.63 + 0.004 * \log W_{vip}^{n} - 0.28 * \log W_{vip}^{n-1} - 0.9 * \log Mr^{n}}$$

$$A_{pb}^{n+1} = A_{pb}^{n} + 74\,172\,373 - 0.6 * A_{pb}^{n} + 0.003 * D_{cl}^{n}$$

$$T^{n+1} = T^{n} + 6\,895\,751\,228 + 0.22 * T^{n} - 0.82 * T^{n-1} + 0.21 * G_{exp}^{n}$$

$$D_{cl}^{n+1} = D_{cl}^{n} - 1\,722\,316\,970 + 0.2 * D_{cl}^{n} + 0.09 * T^{n} \tag{5}$$

According to the F criterion, all seven models are significant. The coefficients of determination of the given models are (sequentially) 72.5%, 56%, 98.2%, 94.6%, 95.9%, 96.4%, and 97.9%. Thus, all the models described are quite effective. Figure 7 shows some graphs where the actual and theoretical dynamics of the studied variables are presented.

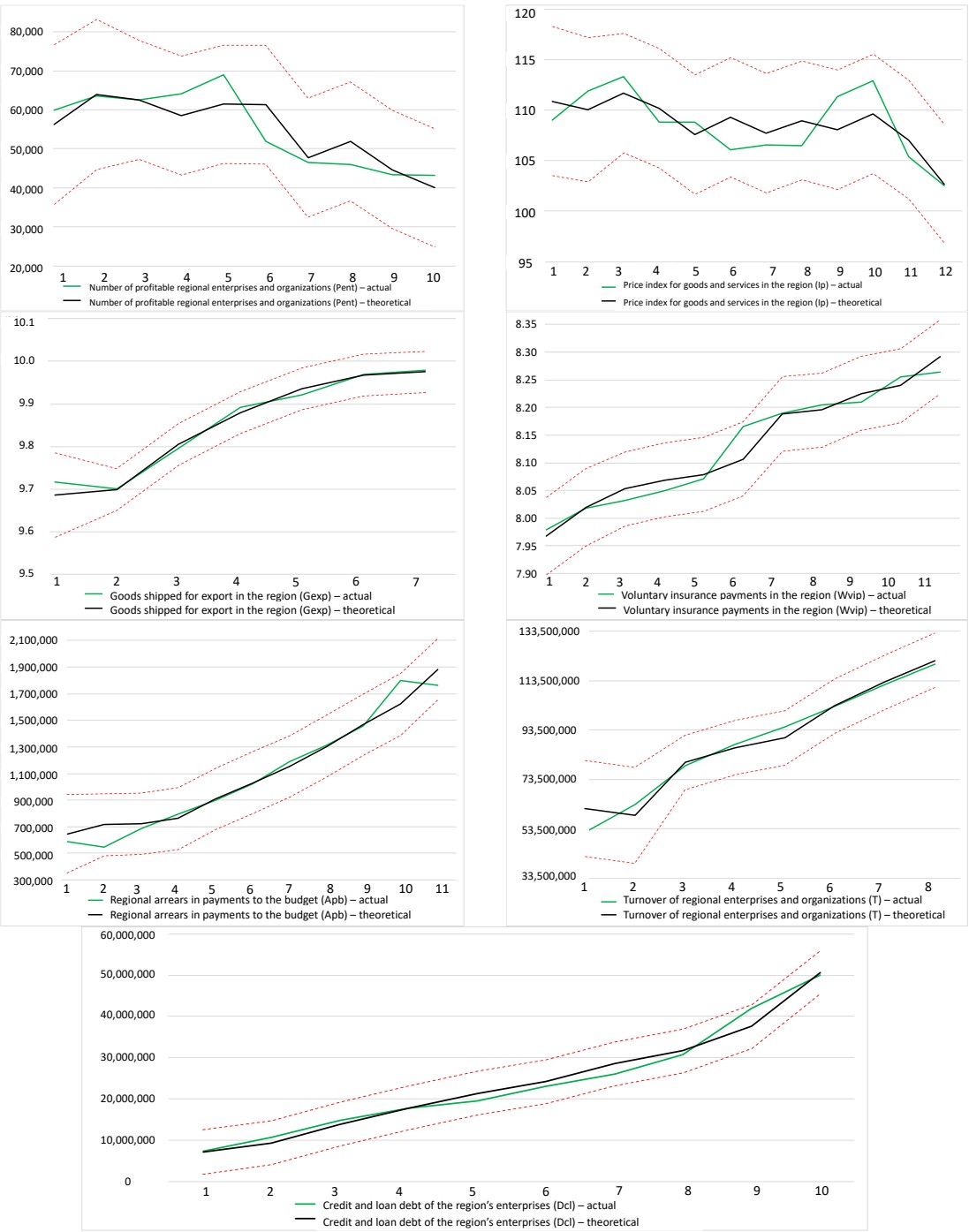

**Figure 7.** Dynamics of the actual and theoretical values of the indicators of the economic environment.

As you can see, there are no significant structural breaks or structural spikes on the graphs. The average percentage of error in the above equations is 7%, 0.7%, 1.08%, 0.26%, 6.6%, 4.5%, and 6.1%, respectively. Therefore, the above models can effectively forecast the corresponding indicators with a lead period equal to 1 year. Thus, the system of equations is effective and can be used to forecast the indicators of the region's economic environment.

Next, let us consider the innovative environment of the region. Being the least extensive, this environment is represented by 2 indicators. The equations used to forecast these indicators are aggregated into system 6.

$$E_{ti}^{n+1} = E_{ti}^n + 2\,202\,617\,805 + 0.96 * MA_{E_{ti}^n} + 0.05 * D_{cl}^n - 2.67 * A_{pb}^n$$

$$N_{apt}^{n+1} = N_{apt}^n + 63.2 + 0.61 * N_{apt}^n - 0.55 * N_{apt}^{n-1} + (3.956\text{E} - 10) * T^n \tag{6}$$

According to the F criterion, the models are significant. The coefficient of determination of the first model is 76%, which is sufficient. The coefficient of determination of the second model is 95.3%. The average error rate is 4.78%. The coefficient of determination of the third model is 63.1%, while the average error rate is 6.29%.

Figure 8 shows the graphs of the actual and theoretical dynamics of the studied variables.

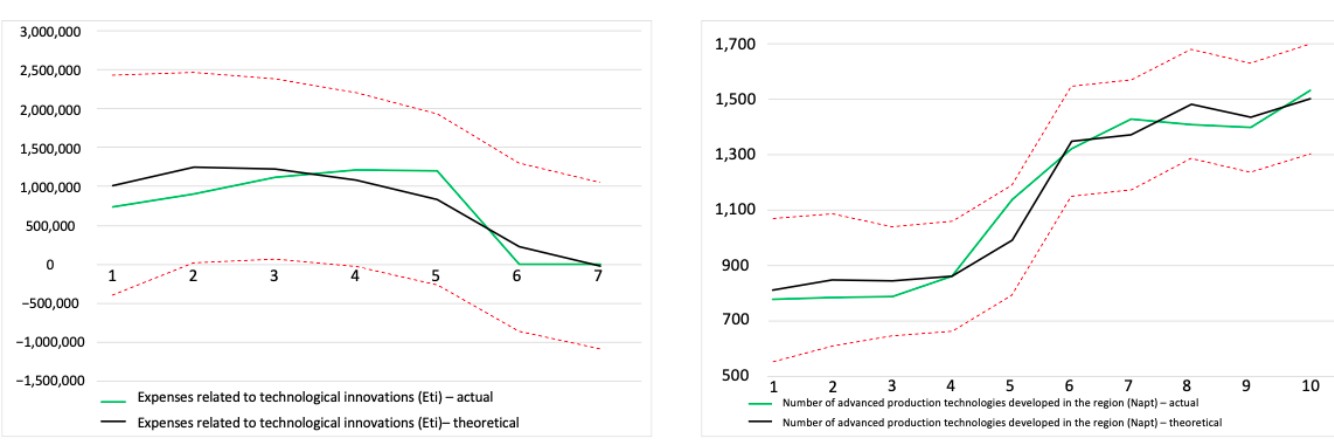

**Figure 8.** The dynamics of the actual and theoretical values of the indicators of the innovative environment.

As you can see, there are no significant structural breaks or structural spikes on the graphs. Thus, the formed system of equations is effective and can be used to forecast the indicators of the region's innovative environment.

In conclusion, let us consider the ecological environment of the region. The equations used to forecast the indicators of this environment are presented in system 7, while Figure 9 shows the graphs of the actual and theoretical dynamics of the studied variables.

$$Mr^{n+1} = Mr^n - 519 - 0.45 * Mr^n + 0.99 * MA_{Mr^n} + 2.38 * C_{hci}^n - 0.001 * N_{apt}^n$$

$$C_{hci}^{n+1} = 213.18 - 0.97 * C_{hci}^n - 0.92 * MA_{C_{hci}^n} - (4.448\text{E} - 10) * E_{lr}^n + 0.0001 * N_{hci}^n$$

$$E_{lrec}^{n+1} = E_{lrec}^n + 1\,318\,158.9 + 0.12 * E_{lrec}^n - 0.56 * E_{lrec}^{n+1} - 0.004 * A_{pb}^n + 0.0001 * D_{cl}^n \tag{7}$$

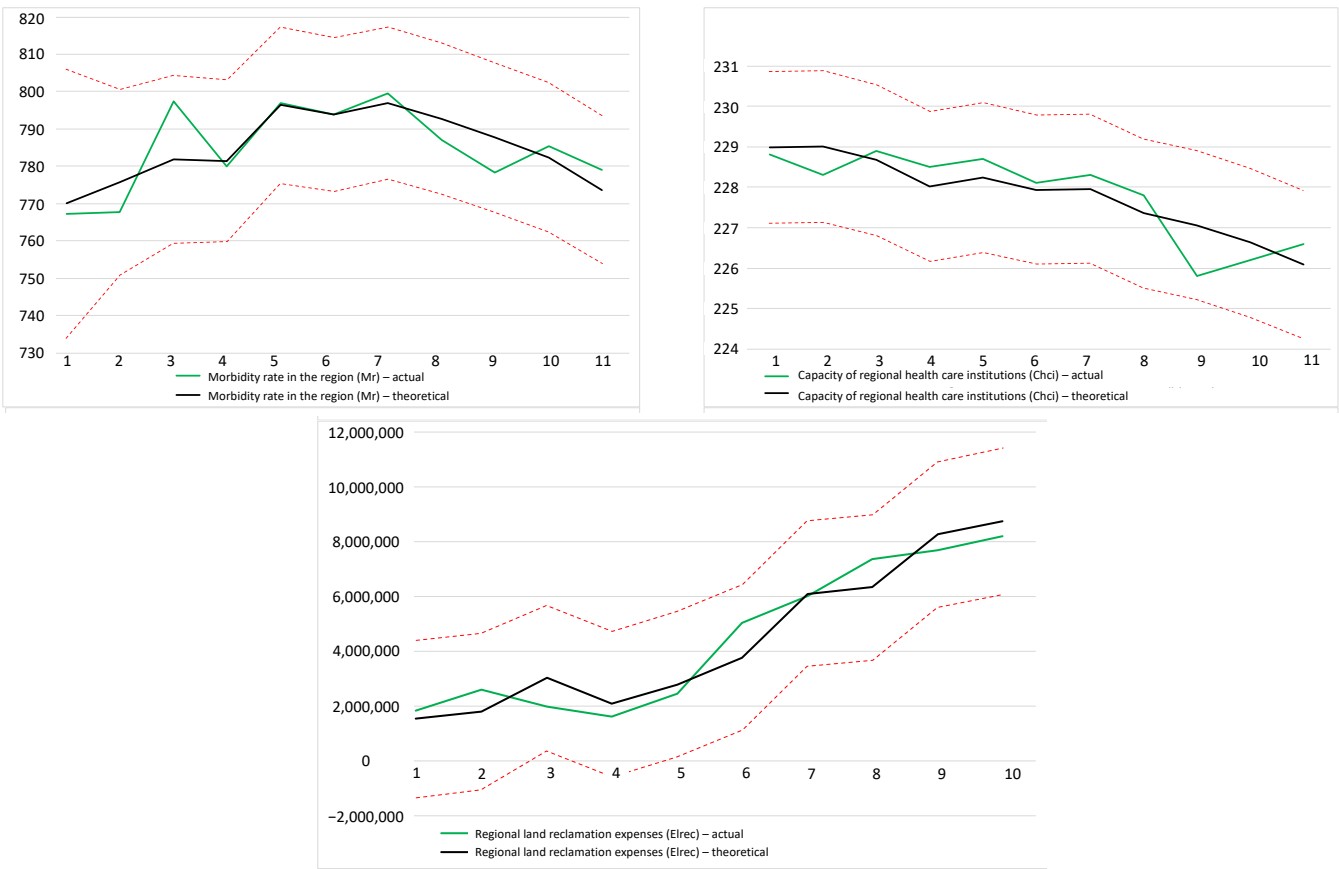

**Figure 9.** Dynamics of the actual and theoretical values of the indicators of the ecological environment.

There are no structural spikes on the graphs. However, there are minor structural breaks in the predicted morbidity rate. This fact is preconditioned by the epidemiological specifics related to forecasting this indicator. Providing the epidemiological environment is stable, this model can be used effectively. The coefficients of determination of the given models are (sequentially) 63.1%, 71.8%, 87.4%, and 91.5%, which confirms the necessary and sufficient quality of the models. The accuracy of the potential forecast is quite high, since the average percentage of error in these equations is 0.63%, 0.21%, 3.5%, and 9%, respectively. Thus, the generated models are effective for predicting the values of the indicators of the region's ecological environment.

At the third stage of the suggested methodology, the development vector of the region's investment attractiveness is valuated. By comparing the current and projected levels, the development vector of the region's investment attractiveness can be determined. This vector can be compared to a conditionally stationary vector ($I_c$), thereby forming a vector model. The degree of differentiation of these vectors is determined by the angle between them. The quantitative value of the width of this angle allows us to assess the dynamics of the development of the region's investment potential. This value can be calculated by determining the inverse cosine of the scalar product of the generated vectors. Figure 10 shows an example of this vector system.

By identifying the angle between the corresponding vectors ($C_d$), the development vector of the region's investment attractiveness can be interpreted:

$$C_d = \arccos\left( \frac{2 \times (t-1) + (I_p - I) \times (I_c - I)}{\sqrt{(t-1)^2 + (I_p - I)^2} \times \sqrt{(t-1)^2 + (I_c - I)^2}} \right) \tag{8}$$

where $t$ is the forecast horizon.

Thus, the region's investment attractiveness is determined using three indicators, namely, the level of the region's investment attractiveness ($I$), the projected level of the region's investment attractiveness ($I_p$), and the development vector of the region's investment attractiveness ($C_d$).

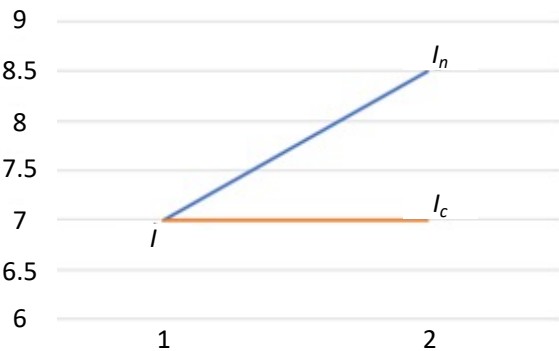

**Figure 10.** Vector interpretation of the change in standardized values.

## 5. Discussion

The tools obtained as a result of the study form a single comprehensive methodology for evaluating the region's investment attractiveness. It should be noted that this methodology is fully consistent with the results of international research studies reviewed above, and combines indicators that consider the region's investment attractiveness in maximum detail. One of the southern regions of Russia—the Republic of Dagestan—is used in order to test this methodology. The indicators were assessed on the basis of open statistics. Figure 11 shows a consolidated graph reflecting the current (as of year 2018) and projected level of the investment attractiveness of the Republic of Dagestan.

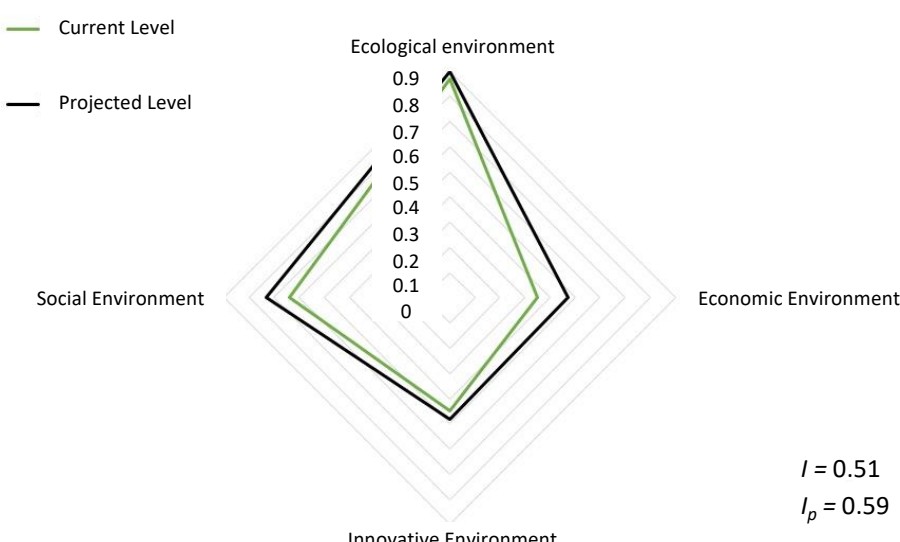

**Figure 11.** Current and projected level of the investment attractiveness of the Republic of Dagestan.

According to the analysis, $I$ is 0.51, which corresponds to subset 3 "Average level of the region's investment attractiveness". As you can see in the graph, this is largely because the values of indicators of the economic environment are low and the values of indicators of the ecological environment are extremely positive. However, the specific weight of indicators of the economic environment is much higher, which ultimately determines the belonging of the Republic of Dagestan to the corresponding subset. Nevertheless, $C_d$ is 21.5°, which is an extremely high value. This development vector is preconditioned by the projected growth of the indicators of the social and economic environment and ensures that the Republic of Dagestan transfers from subset 3 to subset 4 "High level of the region's investment

attractiveness". To a large extent, this growth is caused by a positive trend in the changing indicators of the social and economic environment. In particular, the question is about the unemployment rate, the number of profitable enterprises and organizations, as well as the turnover of enterprises. Thus, despite the average current level of investment attractiveness of the Republic of Dagestan, there is a growing trend. The ecological environment makes a significant contribution to the development of this indicator. However, the innovation environment is stagnating. At the same time, the innovation environment is one of the key mediators of the economic environment. A growth in innovation activity is possible if the share of high-tech enterprises increases, high-tech clusters are created and the number of research centers rises. There is a certain level of dialecticism, since the main sources for the development of the innovation environment are direct investments. Consequently, it can be assumed that at the current level of development of the ecological environment and with the current trends of the social environment, an internal impulse must be created to mobilize reserves that will increase the development level of the innovation environment. In order to undertake the necessary activities, first of all, the basic innovation core must be identified. After that a competitive cluster that is attractive for investment can be set. According to the analysis, the most developed environment in this region is the ecological environment. This fact is largely due to the territorial and climatic specifics of the region. Consequently, the innovative core should be formed with an aim to embrace these competitive qualities, which is possible if specialized research institute are identified and then developed in a systemic way. After systemic interaction between specialized institutions is established, a set of macro-projects can be generated in the field of agriculture. Initial public investment will result in the formation of the necessary infrastructure for systemic interaction and, as a result, a competitive agricultural research cluster can be obtained. After initial macro-projects are implemented, the results obtained can be presented to the world community. In order to increase the investment attractiveness, tax incentives must be introduced for relevant activities in the region. Due to these incentives and the well-developed research infrastructure, it will be possible to attract investments for the development of high-tech agricultural enterprises and thereby increase the flow of investments into the region. This impulse, provided that there is a proper level of performance, can produce a synergy effect that will lead to a huge growth in the investment attractiveness of the Republic of Dagestan. Applying the created model consistently and making conclusions based on the results obtained will help to formulate an algorithm for increasing the investment attractiveness of the Republic of Dagestan.

The key advantages of the research methodology are the quantitative validity of the quality criterion for the resulting models, the unambiguous quantitative nature of the basic variables, which results in the possibility of complete duplication of the research process with obtaining a uniform result, as well as taking into account the dynamics of the innovative development of the region and its impact on the basic variables, which makes it possible to form more reasonable management decisions. The use of this methodology allows the formation of a complex cyclical toolkit.

## 6. Conclusions

This study presents a comprehensive methodology for evaluating the region's investment attractiveness that corresponds to modern digitalization trends. This methodology consists of three key indicators, namely, the level of the region's investment attractiveness ($I$), the projected level of the region's investment attractiveness ($I_p$), and the development vector of the region's investment attractiveness ($C_d$). After analyzing the ratio of these indicators, it is possible not only to make conclusions about the region's investment attractiveness and the development dynamics of this indicator, but also about the ways of increasing both this indicator and the development vectors of business in the studied region. The key difference between the developed methodology and the existing ones is that the new methodology is comprehensive. The methodology is based on a set of indicators which are used to evaluate the economic environment of the region, as well as its

social, innovative, and ecological environments, which fully meets the current trends in the transformation of the criteria applied to evaluating the region's investment attractiveness for business. The key limitation of this study lies in the methodology for collecting statistical information. Statistical information used is collected exclusively by state services on the annual basis and with a delay of several months. At the same time, the level of reliability of the data may differ depending on the region. The potential effect of the use of the developed tools can be significantly increased by clarifying statistical data and increasing the frequency of their determination, which can be implemented exclusively by the federal state statistics service. Also, a minor limitation of the study is the impossibility of taking into account the regional transformation under the influence of the COVID-19 pandemic, since the relevant statistics have not yet been provided at the moment. Consideration of this specificity is possible exclusively at the conceptual level. This paper will be useful, first of all, to the representatives of regional administration, managers of growing business, and to researchers in the field of regional development.

**Author Contributions:** Conceptualization, D.G.R. and M.N.N.; methodology, E.A.K.; formal analysis, E.A.K. and M.N.N.; investigation, M.N.N.; resources, D.G.R.; data curation, M.N.N.; writing—original draft preparation, E.A.K.; project administration, D.G.R.; funding acquisition, D.G.R. All authors have read and agreed to the published version of the manuscript.

**Funding:** This research received no external funding.

**Institutional Review Board Statement:** Not applicable.

**Informed Consent Statement:** Not applicable.

**Data Availability Statement:** Unified Interdepartmental Information and Statistical System (EMISS)—https://fedstat.ru/ (accessed on 10 January 2021).

**Acknowledgments:** This research work was supported by the Academic Excellence Project 5–100 proposed by Peter the Great St. Petersburg Polytechnic University.

**Conflicts of Interest:** The funders had no role in the design of the study; in the collection, analyses, or interpretation of data; in the writing of the manuscript, or in the decision to publish the results.

## Appendix A

**Table A1.** Array of input data 1.

| Definition | Regional Unemployment Indicator | Price Index on the Regional Housing Market | Labor Remuneration Expenses of Regional Enterprises and Organizations | Number of Profitable Regional Enterprises and Organizations |
|---|---|---|---|---|
| Variable Names | UR | $I_{hp}$ | $E_{Ir}$ | $P_{ent}$ |
| Data Source | Unified Interdepartmental Information and Statistical System (EMISS)—https://fedstat.ru/ (accessed on 10 January 2021) | | | |
| Period 1 | 7.45 | 119.28 | 3,239,978,557 | 59,954,00 |
| Period 2 | 6.20 | 105.54 | 4,136,492,347 | 63,518,00 |
| Period 3 | 8.30 | 92.86 | 5,057,736,365 | 62,475,00 |
| Period 4 | 7.30 | 105.70 | 5,105,530,155 | 64,056,00 |
| Period 5 | 6.50 | 108.49 | 5,620,530,892 | 68,899,00 |
| Period 6 | 5.50 | 110.35 | 6,532,338,498 | 51,822,00 |
| Period 7 | 5.50 | 103.64 | 7,145,987,864 | 46,533,00 |
| Period 8 | 5.20 | 104.80 | 7,925,915,422 | 46,039,00 |
| Period 9 | 5.60 | 95.50 | 8,566,343,564 | 43,315,00 |
| Period 10 | 5.50 | 97.12 | 9,139,181,149 | 43,163,00 |

**Table A2.** Array of input data 2.

| Definition | Price Index for Goods and Services in the Region | Goods Shipped for Export in the Region | Voluntary Insurance Payments in the Region | Arrears in Payments to the Budge |
|---|---|---|---|---|
| Variable Names | $I_p$ | $G_{exp}$ | $W_{vip}$ | $A_{pb}$ |
| Data Source | Unified Interdepartmental Information and Statistical System (EMISS)—https://fedstat.ru/ (accessed on 10 January 2021) | | | |
| Period 1 | 109.00 | - | 7.98 | 590,186,355.00 |
| Period 2 | 111.87 | - | 8.02 | 544,352,734.00 |
| Period 3 | 113.28 | - | 8.03 | 686,264,210.00 |
| Period 4 | 108.80 | 9.72 | 8.05 | 793,363,990.50 |
| Period 5 | 108.78 | 9.70 | 8.07 | 900,463,771.00 |
| Period 6 | 106.10 | 9.80 | 8.17 | 1,021,624,187.00 |
| Period 7 | 106.57 | 9.89 | 8.19 | 1,187,276,942.00 |
| Period 8 | 106.47 | 9.92 | 8.20 | 1,312,417,153.00 |
| Period 9 | 111.35 | 9.97 | 8.21 | 1,457,681,943.00 |
| Period 10 | 112.91 | 9.98 | 8.26 | 1,797,765,541.00 |

**Table A3.** Array of input data 3.

| Definition | Turnover of Regional Enterprises and Organizations | Credit and Loan Debt of the Region's Enterprises | Expenses Related to Technological Innovations | Number of Advanced Production Technologies Developed in the Region |
|---|---|---|---|---|
| Variable Names | $T$ | $D_{cl}$ | $E_{ti}$ | $N_{apt}$ |
| Data Source | Unified Interdepartmental Information and Statistical System (EMISS)—https://fedstat.ru/ (accessed on 10 January 2021) | | | |
| Period 1 | - | 7,271,334,250.00 | - | 780.00 |
| Period 2 | - | 10,753,389,124.00 | - | 787.00 |
| Period 3 | 52,218,856,346.00 | 14,826,193,845.00 | - | 789.00 |
| Period 4 | 63,540,559,006.00 | 17,679,954,742.00 | 733,815,967.60 | 864.00 |
| Period 5 | 79,039,407,949.00 | 19,516,948,836.00 | 904,560,846.10 | 1138.00 |
| Period 6 | 87,651,322,102.00 | 23,064,003,158.00 | 1,112,429,217.50 | 1323.00 |
| Period 7 | 94,791,100,235.20 | 25,929,715,547.00 | 1,211,897,098.10 | 1429.00 |
| Period 8 | 102,965,068,304.10 | 30,808,918,599.00 | 1,203,638,084.30 | 1409.00 |
| Period 9 | 111,801,236,272.30 | 42,008,857,928.00 | 1,284,590.33 | 1398.00 |
| Period 10 | 120,158,889,408.50 | 50,089,020,690.00 | 1,404,985.29 | 1534.00 |

**Table A4.** Array of input data 4.

| Definition | Morbidity Rate in the Region | Capacity of Regional Health Care Institutions | Regional Land Reclamation Expenses |
| --- | --- | --- | --- |
| Variable Names | Mr | Chci | Elrec |
| Data Source | Unified Interdepartmental Information and Statistical System (EMISS)—https://fedstat.ru/ (accessed on 10 January 2021) | | |
| Period 1 | - | 7,271,334,250.00 | - |
| Period 2 | - | 10,753,389,124.00 | - |
| Period 3 | 52,218,856,346.00 | 14,826,193,845.00 | - |
| Period 4 | 63,540,559,006.00 | 17,679,954,742.00 | 733,815,967.60 |
| Period 5 | 79,039,407,949.00 | 19,516,948,836.00 | 904,560,846.10 |
| Period 6 | 87,651,322,102.00 | 23,064,003,158.00 | 1,112,429,217.50 |
| Period 7 | 94,791,100,235.20 | 25,929,715,547.00 | 1,211,897,098.10 |
| Period 8 | 102 965,068,304.10 | 30,808,918,599.00 | 1,203,638,084.30 |
| Period 9 | 111,801,236,272.30 | 42,008,857,928.00 | 1,284,590.33 |
| Period 10 | 120,158,889,408.50 | 50,089,020,690.00 | 1,404,985.29 |

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
