# Peer review of "A Transformation of the Approach to Evaluating a Region’s Investment Attractiveness as a Consequence of the COVID-19 Pandemic"

_economies, doi:10.3390/economies9020059_

Round 1

Reviewer 1 Report

The research does not clearly define the hypothesis

When reviewing previous research the authors does not give a critical evaluation of the research

In conclusion authors does not state what limitations were present during the research

It is necessary to state what advantages the methodologies used are in relation to previous research

Author Response

Dear Reviewer, we are very grateful for your comments. Let me give a short response on each of them: 
1.     Indeed, in the process of preparing the research, we excluded a detailed description of hypotheses. Now the description of the hypotheses is added to the text of the article.
2.     Unfortunately, we have excluded some of the critical overview of literature sources in order to reduce the size of the article. In the revised version, we have added such points. 3.     We've also added the description of key limitations.
4.     We noted the advantages of the methodology in the Discussion section of the article.
The attachment contains a revised version of the article. All changes are highlighted in red. We are very grateful to you. Thanks for your comments, after making corrections the article has become much more valuable.

Reviewer 2 Report

1. Motivation for the work should be enhanced in the abstract as well as in the introduction section.

2. More insight discussions in detail based on the concluded advantages shall explored.

Author Response

Dear Reviewer, we are very grateful for your comments. Let me give a short response on each of them:
1.     Unfortunately, we have really shortened the motivation section. In the revised version we have supplemented the abstract and introduction with a detailed description of the motivation.
2.     We have expanded the Discussion section with some advantages of the developed methodology.
The attachment contains a revised version of the article. All changes are highlighted in red. We are very grateful to you. Thanks for your comments, after making corrections the article has become much more valuable.

Reviewer 3 Report

The review of the paper titled “Transformation of the approach to evaluating a region’s investment attractiveness in consequence of the COVID-19 pandemic” submitted to Resources journal:

Major concerns:

  • I would suggest splitting the introductory part of the paper into two sections: (i) introduction, as well as the (ii) literature review on the factors affecting regional attractiveness. In the introduction, one should briefly describe the reasoning for undertaking the research, the state of the research, novelty, brief description of the findings, brief description of implications, the remainder of the paper. The remaining part of the literature review should be moved into the literature review section, adequately named.
  • The author(s) could describe in a more sophisticated manner the role of SEZs in boosting regional attractiveness through tax exemptions, also referencing other research, https://www.ersj.eu/journal/1955/download
  • One could give more clues about why the fuzzy-multiple approach fits best the research problem undertaken in the paper (page 6).
  • The paper lacks discussion with other findings, in the sense of whether your results/methods/implications are in line or against other empirical contributions within the field. Typically, you put discussion after the results.
  • In the concluding section, one should write more about implications stemming from your approach to measuring attractiveness for regional policy.

Minor concerns:

  • Please rewrite the fig. 1 as it is blurry
  • 3, line 133 “McDonald and Bailly в 2017” you should remove the “B” and change the citation style to Oxford
  • 3, line 147, “[fd23, 24].” One should remove “fd”
  • Please verify if Cyrillic characters are present in the paper and change them into Latin ones.

Author Response

Dear Reviewer, we are very grateful for your comments. Let me give a short response on each of them:
1.     The article was originally structured exactly as you suggested. We have restored the original structure and expanded the introduction part.
2.     We have expanded the case for using fuzzy logic.
3.     We have expanded the Discussion section with some advantages of the developed methodology, as well as the literature review section.
4.     We have eliminated all technical issues.
The attachment contains a revised version of the article. All changes are highlighted in red. We are very grateful to you. Thanks for your comments, after making corrections the article has become much more valuable.

Round 2

Reviewer 3 Report

I accept the paper in the present form.

Author Response

Thank you for your review

This manuscript is a resubmission of an earlier submission. The following is a list of the peer review reports and author responses from that submission.